# Sickness Presenteeism as a Link between Long Working Hours and Employees’ Outcomes: Intrinsic and Extrinsic Motivators as Resources

**DOI:** 10.3390/ijerph19042179

**Published:** 2022-02-15

**Authors:** Luo Lu, Cary L. Cooper

**Affiliations:** 1Department of Business Administration, National Taiwan University, Taipei 106, Taiwan; 2Alliance Manchester Business School, University of Manchester, Manchester M15 6PB, UK; cary.cooper@manchester.ac.uk

**Keywords:** presenteeism, work value orientation, well-being, performance, resource loss/gain

## Abstract

The aim of this study was to test the flow from long working hours to sickness presenteeism behavior and its outcomes for employees, while integrating intrinsic and extrinsic work value orientations as moderators in the process. We employed a two-wave design with a five-month interval. Data were obtained from 275 employees in Taiwan. The results of latent moderated structural equations (LMS) revealed that long working hours were positively associated with presenteeism, which in turn was negatively related to employees’ well-being and job performance. Furthermore, the negative indirect effect of working long hours on job performance via presenteeism was weaker for those with a higher intrinsic work value orientation. The negative indirect effect of working long hours on well-being via presenteeism was weaker for those with a higher extrinsic work value orientation. We demonstrated that the long-term impact of presenteeism behavior could be understood by viewing work value orientations as resource gains to compensate resource depletion in a demanding work context. This resource dynamism is pivotal to realizing the functional or dysfunctional outcomes of presenteeism behavior. Theoretical and managerial implications of the findings for employees’ well-being and organizational effectiveness are discussed.

## 1. Introduction

Sickness presenteeism (presenteeism hereafter) refers to situations where people still turn up at their jobs despite ill health that should prompt rest and absence from work [1]. The act of presenteeism due to ill health is the conceptualization we use in the present study. It is problematic for an individual, as it might generate a deterioration of health and well-being [2]; presenteeism also creates costs for organizations and wider society [3]. Thus, presenteeism is often viewed as a negative phenomenon that should be brought under control [4]. Systematic literature reviews and meta-analyses have identified various antecedents of presenteeism and concluded that it is more susceptible to organizational demands (e.g., workload and downsizing) than to personal demands (e.g., financial needs) [1,5,6]. Miraglia and Johns proposed a dual-path meta-analytic model explaining how job insecurity, personal financial difficulties, job control, job demands, collegial support, supervisor support, and optimism as key antecedents affecting job satisfaction and/or personal health, which then lead to sickness presenteeism or absenteeism [1]. Their meta-analytic model, however, stops at the presenteeism decision, not including the consequences of the behavior. Other qualitative reviews mostly relied on evidence from cross-sectional studies to conclude that presenteeism was associated with negative outcomes for employees’ well-being and organizational effectiveness [6,7,8].

Researchers have argued that presenteeism-incurred productivity loss is over-estimated due to conceptual and measurement problems [4,9]. Evidence linking presenteeism to negative outcomes is equivocal in longitudinal studies [10]. For example, although Demerouti, Le Blanc, Bakker, Schaufeli, and Hox found that presenteeism was related to burnout among Dutch nurses [11], Lu, Peng, Lin, and Cooper found no lasting effects of presenteeism on mental health, physical health, or burnout for Chinese employees [12]. Other studies found no evidence of long-lasting damaging effects of presenteeism on productivity and job performance [13,14]. The most recent study from Taiwan found mixed results: presenteeism had a damaging effect on employees’ innovative performance 6 months later, but no such effect was found for burnout [15]. The latest comprehensive review thus concluded that consequences of presenteeism behavior for organizations and individuals are still poorly understood [4].

One way to reconcile the inconsistencies is to look for potential mechanisms that might affect the link between presenteeism and its consequences [4,8,10]. It is intriguing that factors reflecting positive motives such as job satisfaction, organizational commitment, and work engagement have been consistently linked to presenteeism [1,16], the behavior that is often portrayed as aversive. However, motivational factors are under-studied in the presenteeism research [4,10]. To address this void, Ma, Meltzer, Yang, and Liu differentiated between autonomous and controlled motivation for presenteeism [17]. Cooper and Lu proposed a comprehensive theoretical framework to explain how intrinsic or extrinsic work value orientations motivate an individual to commit presenteeism behavior voluntarily or involuntarily, which then leads to good or bad outcomes pertaining to well-being and productivity [10]. The central tenet of Cooper and Lu’s motivational theory of excessive availability for work (EAW) is that the act of excessive working (e.g., presenteeism) per se is neutral; it is the configuration of personal, organizational, and cultural factors that determines outcomes for individuals and organizations [10]. Taking the same motivational theoretical perspective, while Cooper and Lu’s model views intrinsic and extrinsic work value orientations as motivational *antecedents* manifested as different types of driving forces (i.e., voluntarily or involuntarily) for committing presenteeism behavior, we in the present study attempted to unpack the *mechanisms* of work value orientations on the “presenteeism–outcomes” link. Specifically, we empirically tested the resource loss/gain dynamism postulated in the conservation of resource (COR) theory [18] to examine whether resource gains (i.e., intrinsic/extrinsic drives to accomplish work) can protect an individual’s well-being and task performance in the context of resource depletion (i.e., working when ill due to heavy workload) [19].

For this purpose we drew on the core COR proposition of resource loss to theorize that working long hours due to a heavy workload can result in presenteeism behavior, which further depletes resources and leads to negative consequences for well-being and job performance over time. As an antidote, critical resource gains, such as the motivational drive for hard working (i.e., intrinsic and extrinsic work value orientations), may impact the final consequences for an individual [10]. The ultimate goal was to arrive at a more comprehensive understanding of the chain of events, revealing how long work hours precipitate presenteeism behavior, which leads to deleterious consequences for well-being and productivity over time, as well as how personal motivational resources may alleviate such an impact.

## 2. Theoretical Framework and Hypotheses Development

### 2.1. Resource Loss Cycles in the “Long Working Hours–Presenteeism–Outcomes” Linkage

*Long working hours, presenteeism, and COR*. The basic tenet of COR theory is that people strive to retain, protect, and build resources; moreover, the potential or actual loss of these valued resources threatens or is stressful for individuals [18]. Resource loss is thus central to the stress and adaptation experience. Moreover, the loss of resources tends to lead to resource loss cycles that have increasing strength and speed. Working extended hours, often resulting from overwhelming workloads, precipitates the likelihood for working while sick to avoid the piling up of work tasks [1]. Viewed from the COR perspective, sickness presenteeism thus represents a scenario for resource depletion [20]. That is, when employees strive to accomplish work demands under suboptimal health states, the continuous resource depletion triggers the resource loss cycle.

Systematic reviews and the latest work on presenteeism have revealed that the majority of research on *antecedents* focused on quantitative, emotional, social, cognitive, and physical job demands [1,4,8,21,22,23], while prolonged working hours were largely overlooked in this context [24]. However, the heightened pressure to work longer hours has become a salient work condition worldwide as organizations strive to do more with less, compounded by the devastating disruptions and uncertainties caused by the COVID-19 pandemic [10,15,25]. Extending previous studies, we thus focused on long working hours as a salient aspect of work conditions for Chinese employees in examining the resource loss cycle triggered by presenteeism.

*Long working hours and employees’ outcomes.* In line with the COR theory, working for an extended period of time requires protracted, energy-draining efforts, which may overtax an individual’s resources and impair well-being [8]. We further argue that having to work long hours would deprive employees of the opportunity to recuperate and restore energy. Such continuous resource depletion situations exacerbate the risk for illness and precipitate the propensity of employees to work on days when they have suboptimal health to avoid the piling up of unfinished tasks [10]. A longitudinal study in Taiwan found that working long hours was positively associated with employees’ exhaustion and negatively associated with work engagement and job performance [26]. We therefore propose that working long hours will damage employees’ quality of work life (i.e., well-being and job performance).

**Hypothesis** **1a** **(H1a).**
*Long working hours are negatively associated with well-being.*


**Hypothesis** **1b** **(H1b).**
*Long working hours are negatively associated with job performance.*


*The mediating role of presenteeism.* The upswing in working hours is evidently more severe in some East Asian countries due to weaker employment legislation and regulations [27]. The “long-working-hour norm” is sanctioned by the deeply rooted hardworking cultural values of Confucian ethics [28]. These collectively held moral imperatives sustain the long-existing practice of prolonged work hours in countries such as Japan, Taiwan, Hong Kong, Singapore, and Korea [24,29,30]. In Taiwan there is no mandatory ceiling on working hours for most white-collar professionals (i.e., so-called “job responsibility system”), and working long hours is a widely used image-management tactic among office workers [24]. One recent study found that white-collar workers, on average, worked over 46 h per week in Taiwan [26]. Findings from the latest qualitative and crosscultural survey-based research corroborate the postulation that work norms and social doctrines affect presenteeism behavior [31,32]. Recent cross-sectional studies in Australia [22,23] also found that higher job demands were positively related to burnout, which then leads to presenteeism. We extended the existing studies to examine the lasting effects of long working hours as a job demand on well-being and job performance via presenteeism behavior.

While presenteeism may seem attractive for organizations at a first glance (avoiding productivity loss due to sickness absence), employers are now realizing that it may incur hidden productivity costs in addition to the more obvious wear and tear on employees’ well-being. This is because when employees work in suboptimal health energy is sapped and attention diverted, such that productivity and performance quality (e.g., innovation) suffer [15,33]. In other words, heavy workloads tend to compel employees to commit presenteeism behavior, which over an extended period of time can lead to exhaustion, ill-being, and productivity damage in the ensuing resource loss cycle [19]. Therefore, we hypothesized:

**Hypothesis** **2a** **(H2a).**
*Long working hours will be positively related to presenteeism, which in turn will lead to a decrease in well-being.*


**Hypothesis** **2b** **(H2b).***Long working hours will be positively related to presenteeism, which in turn will lead to a decrease in job performance*.

### 2.2. Work Value Orientations as Resource Gains in the “Long Working Hours–Presenteeism–Outcomes” Process

Existing cross-sectional studies have largely confirmed the damaging effects of presenteeism on well-being and productivity. However, the findings from the handful of longitudinal studies are inconsistent, suggesting that the consequences of presenteeism behavior may vary when assessed in an extended timeframe [10]. One possibility for such heterogeneity is that some potent individual difference factors may act as buffers to weaken the noxious effects of presenteeism. We thus focused on work value orientations as motivational forces to investigate the moderating mechanisms in a longitudinal design in the integrated “long working hours–presenteeism–outcomes” process.

*Work value orientations.* Work value orientation refers to work-related reinforcement preferences or tendencies to value specific types of incentives in the work environment [34]. People vary on the relative importance they attach to one versus another type of value, such that some people give greater weight to extrinsic values and others to intrinsic values [35]. Organizational researchers have distinguished intrinsic from extrinsic work value orientations [36,37,38]. An intrinsic work value orientation is conceptualized as encompassing factors that are intrinsic to the job (e.g., autonomy, competence, interesting, and challenging). An extrinsic work value orientation is conceptualized as referring to financial and social rewards as well as security offered by a job. Numerous empirical studies and meta-analyses have revealed that an emphasis on intrinsic goals, relative to extrinsic goals, is associated with better well-being and task performance [36,39,40].

*Work value orientations and the motivational mechanism of resource gains.* Responding to Ruhle et al.’s observation that employees’ positive motives are understudied yet important mechanisms for understanding presenteeism [4], we contend that employees who have a high intrinsic work value orientation may consider presenteeism as a means to exercise autonomy, master competence, and receive satisfaction from interesting and challenging work [10]. Research on presenteeism has found that in the challenging long working hours of the Chinese work setting, driven by intrinsic motivation, some employees voluntarily go to work when ill because they truly enjoy the job and want to show diligence and loyalty [41]. Thus, when they commit presenteeism they will exert more effort to compensate for potential performance decrements due to illness [42]. Through such motivational self-regulation the detrimental effects of presenteeism on job performance and well-being may be alleviated.

On the contrary, extrinsically oriented people primarily focus on obtaining external indicators of worth, such as social approval and rewards. Two meta-analyses of organizational research have confirmed that providing financial incentives is associated with higher performance [43,44]. A subsequent literature review further presented empirical evidence for the proposition that extrinsic incentives are as beneficial as intrinsic rewards in realizing high job performance [37]. Research on presenteeism has also found that in the challenging long working hours of the Chinese work setting, driven by extrinsic motivation, some employees go to work when ill because they want to protect their income and job security, gain social approval from significant others (supervisors, colleagues, and customers), secure career prospects, and show team spirit [41]. Through such motivational self-regulation the detrimental effects of presenteeism on job performance and well-being may be alleviated.

In the context of the present study, for Taiwanese employees working under the demanding conditions of long hours and through illness, resource depletion is high and continuous, amplifying the value of resource gains [19]. Thus, intrinsic and extrinsic work value orientations may act as personal motivational resources of “drives within” for employees to construct meaning and purpose with which to protect well-being and persevere in maintaining performance standards. Accordingly, we proposed that work value orientations would moderate the indirect effect of long working hours on well-being and job performance via presenteeism behavior. We thus hypothesized:

**Hypothesis** **3a** **(H3a).***Intrinsic work value orientation will moderate the indirect effects of long working hours on well-being and job performance* via *presenteeism, such that the effect*
*s will be weaker for employees with a higher intrinsic work value orientation.*

**Hypothesis** **3b** **(H3b).***Extrinsic work value orientation will moderate the indirect effects of long working hours on well-being and job performance* via *presenteeism, such that the effect**s will be weaker for employees with a higher extrinsic work value orientation.*

Figure 1 depicts the research models presented above, one with an intrinsic work value orientation and the other with an extrinsic work value orientation as the moderator.

## 3. Method

### 3.1. Procedure and Participants

All of our participants were white-collar full-time employees working in different organizations of diverse industries across Taiwan. We employed a two-wave study design in which all independent variables (working hours, presenteeism, and intrinsic as well as extrinsic work value orientations) were measured at time 1 (T1) and dependent variables (well-being and job performance) were measured twice with an interval of five months. In all of the analysis reported below well-being and job performance measured at time 2 (T2) were used as dependent variables, while those at T1 were treated as the baseline levels and controlled for. While there is a constant call for more longitudinal studies in organizational research, there is no consensus for the optimal temporal frame [45]. The Demerouti et al. study found a long-term effect (time frame of 1.5 years) of presenteeism on burnout for Dutch nurses [11], and the two Taiwanese studies found a short-term effect (time frames of 2 and 3 months) of presenteeism on well-being and job satisfaction [12,41]. However, neither of the Taiwanese studies found a significant relationship between presenteeism and job performance. We thus decided to test the effects of working hours and presenteeism on well-being and job performance in a medium-term timeframe (5 months), allowing sufficient time for presenteeism to incubate its effects on job performance and well-being.

The present study was approved by the Research Ethics Committee of the principal researcher’s institute. A paper–pencil survey was carried out using convenient sampling to recruit participants through personal contacts of the principle researcher. Some participants were enrolled in executive education programs, and others were recruited through managers in various organizations. With the assistance of the contact persons we successfully collected data at two time points. At time 1 (T1) a cover letter accompanied the questionnaire, explaining the aim of our study and assuring confidentiality. The initial survey was completed by 474 persons (response rate: 80.75%). Five months later 291 persons completed the second survey (retention rate of 61.39%). Using respondents’ self-generated “matching” codes, T1 and T2 data from 275 persons were combined. We examined the attrition bias by comparing the participants in the panel sample and the dropouts with regard to demographic characteristics and the mean scores of all variables (T1). We found no significant differences in any variables, indicating no serious attrition bias.

The sample was 36.3% male and 63.7% female, with a mean age of 36.44 (SD = 8.68, range = 23–63) and a mean job tenure of 6.86 years (SD = 6.79). Just over half of the sample (50.2%) were married. Most participants (96.50%) had a college education and a quarter of the respondents (24.70%) were managers. We asked participants to report the size of their organizations in three categories, namely SMEs employing under 250 people (35%), medium enterprises with between 251 and 1000 (27.5%) employees, and large companies employing over 1000 employees (37.50%). We also asked participants to identify the industries of their organizations and found that manufacturing (27.8%), high-tech (22.9%), and service (15.9%) were the top three.

### 3.2. Measures

The structured questionnaire was written in Chinese, and all the standard measures have been used and validated with Chinese samples in previous studies (the Chinese validation reference is given for each scale below).

*Work**ing hours.* Following the practice of a global survey of working hours (e.g., International Labour Organization, OECD) and the suggestion from a literature review on workaholism per working long hours, we assessed weekly work hours in the present study [26,46]. Participants reported their actual working hours in a typical week, including reported as well as nonreported overtime, and working on side jobs.

*Presenteeism.* As pointed out in the latest literature review, studies focusing on presenteeism as a behavior mainly draw upon unvalidated single items [4]. We thus used the only non-single-item presenteeism behavior scale reviewed in Ruhle et al. (Table 2, p. 349). The scale is developed and validated for Chinese populations by Lu et al. to measure the act of “sickness presenteeism” without evaluation or negative connotations (e.g., “Although you felt sick, you still forced yourself to go to work”) [12,13]. With a timeframe of the “past 6 months”, four-point scales were used (1 = never, 4 = more than five times) to rate the behavioral frequency of presenteeism. This scale has shown high reliability and construct validity in Chinese samples [12,13,15,47]. The internal consistency reliability of the scale was α = 0.88 in the present study.

*Job performance.* We used the four-item scale developed by Ang, Van Dyne, and Begley to assess job performance (e.g., “My supervisor is satisfied with the level of my performance”) [48,49]. Five-point rating scales were used (1 = disagree very much, 5 = agree very much). This scale has shown high reliability and validity for Chinese employees [26]. (T1 α = 0.88, T2 α = 0.89.)

*Well-being.* As suggested by Ruhle et al. in the latest literature review, presenteeism behavior may be more related to general well-being than specific health consequences [4]. One previous study measuring presenteeism behavior three months apart from self-rated symptoms failed to find a relationship [12]. We therefore focused on subjective well-being [50], measured with the five-item version of the Chinese Happiness Inventory developed by Lu [51]. The scale has robust reliability and validity for Chinese populations [52]. A sample item is “I am satisfied with most things in my life.” Respondents rated each item on a four-point scale (0 = strongly disagree, 3 = strongly agree). A higher total score indicated a higher level of happiness. (T1 α = 0.89, T2 α = 0.89.)

*Work value orientations.* Intrinsic and extrinsic work value orientations were measured with the ten-item Chinese Work Value Scale [53]. Six items are for intrinsic work value orientation (e.g., “A sense of achievement”, “Challenging work”) and four are for extrinsic work value orientation (e.g., “Good income”, “Job security”). Six-point rating scales were used (1 = disagree very much, 6 = agree very much). The internal consistency reliability was α = 0.87 for intrinsic work value orientation and α = 0.86 for extrinsic work value orientation in the present study.

In addition, information on sex (coded male = 0, female = 1), age, marital status (coded married = 1, not married = 0), education attainment (converted to years of education), tenure of current job (in years), and job position (coded managers = 1, employees = 0) were recorded. Finally, to detect self-report bias we assessed social desirability with 3 items (e.g., “There have been occasions when I took advantage of someone”, negation indicates higher social desirability bias) [54]. Pearson correlation analysis showed that the social desirability score had no significant correlation with any of the main study variables. Similar patterns of results were obtained with or without social desirability as the control variable. For parsimony we report the results below, without controlling for the social desirability score.

## 4. Results

### 4.1. Correlations among Variables

The bivariable correlations were computed, and the results are shown in Table 1. Working hours positively correlated with presenteeism and negatively correlated with job performance in addition to well-being. The average working hours were 44.54 per week, much longer than the statutory cap of 40 h/week legislated in Taiwan.

### 4.2. Primary Analyses

*Measurement model.* Prior to testing the structural relationships we conducted confirmatory factor analyses (CFAs) on T1 and T2 multi-item measures to examine the discriminant validity of our research variables. As shown in Table 2, the hypothesized three-factor model for T1 variables (i.e., intrinsic work value orientation, extrinsic work value orientation, and presenteeism) and the hypothesized two-factor model for T2 variables (i.e., well-being and job performance), respectively, had the best fit relative to the alternative models, indicating the unidimensionality of the measures. Additionally, all factor loadings for the measured variables on their latent factors were significant (presenteeism λ range = 0.68 to 0.76; intrinsic work value orientation λ range = 0.68 to 0.89; extrinsic work value orientation λ range = 0.77 to 0.87; well-being λ range = 0.84 to 0.92; and job performance λ range = 0.81 to 0.88). Furthermore, the composite reliability of the latent factors ranged from ρ = 0.73 to ρ = 0.92 (intrinsic work value orientation ρ = 0.73, extrinsic work value orientation ρ = 0.83, well-being ρ = 0.92, and job performance ρ = 0.89). The results indicate that our measures had adequate construct validity.

### 4.3. Hypotheses Testing

*Testing main effects and mediating effects.* We conducted a path analysis with Mplus (version 6) to test our hypotheses [55]. All manifest variables were standardized prior to the analysis to put them on a common scale. Path estimates and conditional indirect effects were evaluated using bias-corrected confidence intervals based on 2000 bootstrap resamples with replacement [56].

Hypothesis one (a and b) proposed the negative relationships between long working hours and outcomes (well-being and job performance). The positive relationship between the intrinsic and extrinsic work orientations (r = 0.41, Table 1) might be due to the importance that employees attach to work values in general [57]. Thus, to examine the “net effect” of each work value orientation we controlled for the extrinsic orientation when analyzing the effect of intrinsic orientation, and vice versa.

As shown in Table 3, working hours at T1 were negatively associated with well-being (β = −0.102, se = 0.05, and *p* < 0.01) and job performance (β = −0.134, se = 0.04, and *p* < 0.01) at T2, supporting hypotheses 1a and 1b. Table 3 also presents the mediating effects of presenteeism: (1) the indirect effect of working hours on well-being through presenteeism was −0.14 (*p* < 0.005; 95% CI (−0.1013, −0.0018)); (2) the indirect effect of working hours on job performance through presenteeism was -0.21 (*p* < 0.001; 95% CI (−0.1207, −0.0110)). The mediation effects of presenteeism on both the “working hours–well-being” and “working hours–performance” paths were significant. To further evaluate the effect size of the mediation effects we used Preacher and Kelley’s recommended kappa-squared value (κ^2^) [58]. We found that: (1) the effect size for the mediating effect of presenteeism on the relationship between working hours and well-being was medium in strength (κ^2^ = 0.11); (b) the effect size for the mediating effect of presenteeism on the relationship between working hours and job performance was large in strength (κ^2^ = 0.28) [58]. The above results provide full support for hypotheses 1a and 1b.

*Testing the latent moderated mediation effects.* We followed Cheung and Lau’s suggestion to conduct latent moderated structural equation modeling (LMS) in Mplus [59]. We formulated four *latent moderated mediation models* (models three, four, five, and six in Table 4) to examine the moderating effects of intrinsic/extrinsic work value orientations on the indirect effects of working hours via presenteeism on well-being and job performance, respectively (see Figure 1). These four hypothesized moderated mediation models were compared against two base models (models one and two), each with one focal dependent variable. Specifically, model one was the *simple mediation model* linking long working hours to well-being through presenteeism. Model two was the *simple mediation model* linking long working hours to job performance through presenteeism. Model three was the moderated mediation model with *intrinsic work value orientation* affecting the indirect effect of long working hours on *well-being* through presenteeism. Model four was the moderated mediation model with *intrinsic work value orientation* affecting the indirect effect of long working hours on *job performance* through presenteeism. Model five was the moderated mediation model with *extrinsic work value orientation* affecting the indirect effect of long working hours on *well-being* through presenteeism. Model six was the moderated mediation model with *extrinsic work value orientation* affecting the indirect effect of long working hours on *job performance* through presenteeism. In other words, in the latent moderated mediation models (model three to model six) we *added* the interactions among latent predictor variables (i.e., presenteeism × intrinsic/extrinsic work value orientations) to the mediation structural models (i.e., “working hours→presenteeism→well-being/job performance”, model one and model two). As shown in Table 4, the moderated mediation models for well-being (model three and model five) had a better fit than the simple mediation model (base model one). Similarly, the moderated mediation models for job performance (model four and model six) had a better fit than the simple mediation model (base model two). These results indicated the improvement values for including the moderators in the overall structural models. Table 4 also showed that only the two latent interaction terms, namely the presenteeism × intrinsic orientation (on job performance in model four, β = −0.138, *p* < 0.01) and presenteeism × extrinsic orientation (on well-being in model five, β = 0.162, *p* < 0.01), were significant. However, the other two latent interaction terms, namely the presenteeism × intrinsic orientation (on well-being in model three) and presenteeism × extrinsic orientation (on job performance in model six) were not statistically significant. Thus, the latent moderated mediation analysis results *partially* supported H3a and H3b. Specifically, intrinsic work value orientation moderated the indirect effect of long working hours on job performance via presenteeism, whereas extrinsic work value orientation moderated the indirect effect of long working hours on well-being via presenteeism.

To further probe the statistically significant conditional indirect (moderated mediation) effect we used estimates from the latent variable moderated mediation model to calculate the conditional indirect effect of working hours via presenteeism on well-being/job performance at various levels of intrinsic/extrinsic orientation. We estimated the indirect effects at low (−1 SD), medium (1 SD), and high (+1 SD) levels of the moderator. We explored the shape of the significant interactions by conducting a simple slope test [60]. For job performance, the slope of presenteeism for employees with a high intrinsic work value orientation was non-significant (effect = 0.0112, bias-corrected bootstrap confidence interval, BCCI, = (−0.0014 to 0.0005)), whereas the slope of presenteeism for those with medium and low intrinsic orientations were significantly negative (medium: effect = −0.0028, BCCI = (−0.0252 to −0.0134); low: effect = −0.0143, BCCI = (−0.1231 to −0.0623)). This indicates that over a five-month period the relationship between presenteeism and job performance was negative for employees of medium and low levels of intrinsic work value orientation; presenteeism was unrelated to job performance when employees had a high level of intrinsic orientation (see Figure 2). Thus, the results partially supported H3a, that intrinsic work value orientation moderated the indirect effect of long working hours on job performance via presenteeism, such that the effect became nonsignificant for employees with a higher intrinsic work value orientation.

For well-being, the slope of presenteeism for employees of a low extrinsic work value orientation was significantly negative (effect = −0.0113, BCCI = (−0.0096 to −0.0029)), whereas the slope of presenteeism for those of a high extrinsic orientation was significantly positive (effect = 0.0088, BCCI = (0.0023 to 0.0036)). This indicates that over a five-month period the relationship between presenteeism and well-being was negative for employees of a low level of extrinsic work value orientation, but positive for those of a high level of extrinsic orientation (see Figure 3). Thus, the results partially supported H3b, that extrinsic work value orientation moderated the indirect effect of long working hours on well-being via presenteeism, such that the effect was positive for employees with a higher intrinsic work value orientation.

## 5. Discussion

### 5.1. Theoretical Implications

As repeatedly found in reviews of the presenteeism literature, the bulk of existing empirical work in organizational research has focused on categorizing the determinants (or antecedents) of presenteeism, namely understanding the decision process of employees to be present or absent during sickness [1,4,5,8]. Only recently have some theoretical efforts been invested in clarifying and understanding the potential outcomes of presenteeism [10,61,62]. Thus, the conclusion of an earlier review, that “the current body of scientific literature does not provide sufficient evidence to draw conclusions on the consequences of sickness presence” (p. 216), still largely holds [63]. More importantly, we have little insight into the psychological mechanisms, contextual factors, and processes that link presenteeism behavior to its outcomes for individuals and organizations, regarding acclaimed well-being damage and productivity loss [4,10,62,64].

To address this void in knowledge we in the present study examined the complete process of “antecedents–presenteeism–outcomes”, incorporating motivational personal resources as the explanatory psychological mechanisms for affecting the outcomes five months later. More importantly, to answer the call for theoretical developments in presenteeism research [4,8] we pivoted our theoretical framework on the resource loss/gain dynamism of COR while adopting a motivational perspective on presenteeism [10,17,62]. Our results demonstrated that long working hours precipitated employees to work when sick, which led to impaired performance and depressed well-being, thus raising a red flag for the accumulating risk of presenteeism (in a medium-term timeframe). These findings corroborate the resource loss cycle posited in the COR theory by showing that enduring resource depletion caused by long work hours and sickness presenteeism can harm employees’ job performance and well-being. However, intrinsic and extrinsic work value orientations can function as protective mechanisms, setting in motion resource gain cycles to counteract resource loss.

As Cooper and Lu purport in their motivational theory of excessive availability for work (EAW, e.g., sickness presenteeism), that “Good outcomes (work well-being, unharmed work performance) will be more likely among people who commit EAW as a voluntary act through freewill or through internalization” (proposition 4a, p. 9; *italics added*), our findings on work value orientations lend some support for this proposition. Employees who have a high endorsement of intrinsic work value orientation are more likely to commit presenteeism as a voluntary act through freewill, pursuing fun, interests, and challenges at work [1,17,41]. Those who have a high endorsement of extrinsic work value orientation are more likely to commit presenteeism as a voluntary act through internalization, winning rewards and approval by serving one’s duty, and securing career prospects through compliance with norms [13,28].

Corroborating Cooper and Lu’s proposition 4a, we have unraveled the motivational mechanisms of human agency by demonstrating the moderating effects of intrinsic and extrinsic work value orientations on the presenteeism–outcomes linkage [10]. The buffering effects of work value orientation underlined the possibility that with the right configuration of personal, organizational, and cultural factors, presenteeism could have positive outcomes for employees and organizations [4]. Our results highlighted the critical importance of motivational resources in understanding the functional aspect of presenteeism [4,10,17,62].

The protective mechanism of work value orientation can be explained by the satisfaction of basic psychological needs. Corroborating the existing research, individuals with a high intrinsic work value orientation cherish opportunities for intellectual fulfilment, creative self-expression, and the pleasure associated with task mastery on the job [65]. These employees view presenteeism as means to exercise autonomy and to master competence. Such positive motives thus act as personal resources attenuating the negative impact of presenteeism on job performance, as we predicted. Individuals with a high extrinsic work value orientation identify and internalize extrinsic values of monetary rewards, promotion, social recognition, realizing self-worth, and security [36,37]. Such positive motives may enhance self-affirmation and uplift well-being [52], but may not necessarily help performance quality [36]. Our finding that extrinsic work value orientation buffered the negative effects of presenteeism on well-being can also be explained by the effort–reward imbalance (ERI) model [66]. Research on the ERI has robustly found that rewards of esteem, security, pay, and promotion counteract work stress and strain by protecting well-being and health [67].

The intriguing protective effect of extrinsic work value orientation for employees’ well-being during presenteeism pivots on the cultural context of presenteeism [4,10]. Traditional Confucian values put a great emphasis on acquiring stability, security, wealth, and status for the prosperity and welfare of the family [28,68]. In contemporary Chinese societies, aspiring to realize extrinsic rewards offered by a job is thus not only meaningful and essential but also recognized as abiding by the cultural imperative. For employees in a collectivist Chinese society, committing presenteeism to protect their career image satisfies the basic psychological needs for relatedness and competence. Taken together, our findings added intrinsic and extrinsic work value orientations to the list of psychological mechanisms (e.g., self-efficacy, psychological detachment) that can ameliorate the problematic consequences of presenteeism behavior on both well-being and productivity [12,26].

Last but not the least, our findings testified the resource loss/gain dynamism posited in the COR theory by showing that under intensive work situations (i.e., presenteeism triggered by long working hours) personally meaningful motivational resources, whether of an intrinsic or extrinsic source, are effective buffers, alleviating the accumulating deleterious effects of presenteeism on well-being and productivity.

### 5.2. Managerial Implications

What we have found in the present study has implications for both organizational strategies and managerial practices. As we have demonstrated, long working hours had a catalytic effect on sickness presenteeism and its subsequent damaging effects on employees’ job performance and well-being. In our data the average working week was ca. 45 h, comparable to previous findings in Taiwan [26]. It is thus imperative that organizations endeavor to shorten *actual* work hours in order to stamp the origin of the “accumulative consequences of downstream health” [5]. In addition, organizations and managers should reinforce the legitimacy of taking sick leave when needed and adjust task allocation or provide job replacement arrangements to lessen the felt pressure of employees to commit presenteeism. The appropriate use of sick leave and recuperation as a health-promoting strategy is well-documented in the recovery research [69].

Moreover, organizations should create work environments that foster intrinsic work motivation, for example, equipping employees with autonomy and flexibility at work. Organizations should also provide attractive monetary or other extrinsic reinforcers (e.g., praise) that can enhance employees’ perceptions of fairness and organizational support [70]. Such psychological states enhance employee well-being [71]. As shown in the present study and the ERI literature, extrinsic rewards help lift morale and spirits, which keep people going in demanding and precarious work conditions, at least in the short-run [66,67].

### 5.3. Limitations and Directions for Future Research

Our study has some limitations. Firstly, we used self-report measures, which may increase the threat of common method variance (CMV) bias [72]. In an effort to minimize such bias we adopted a longitudinal design to separate the independent variables (working hours, presenteeism, and work value orientations) from dependent variables (job performance and well-being) in time. Furthermore, researchers have noted that CMV might enhance main effects, but that interaction effects were hardly attributable to such method bias [72]. Thus, it is unlikely that our evidence for a moderated mediation process is solely due to common method bias. However, future research may adopt a supervisor–employee dyadic design to crossvalidate our findings by obtaining supervisor-rated job performance data.

We extended the studies on presenteeism to an East Asian society (Taiwan); however, the relatively small and nonrandom sample we had may limit the generalization of our findings. Future studies should include more diverse samples from other Asian countries to establish the generalizability of our findings as well as those from Western studies. In addition, a lack of firm-level controls may be an omitted variable bias. Previous studies on presenteeism have highlighted the relevance of occupations, for instance in terms of the personal obligations teachers and care workers feel towards pupils and care recipients [4]. We probed some of the organizational factors in the present study, such as the organizational climate of presenteeism and job replacement policy [32]; these results were not reported in the tables as they were non-significant. Nonetheless, future research could target specific occupations, such as professionals and service workers, to better contextualize the study of presenteeism.

Finally, we measured the effects of presenteeism on well-being and performance after 5 months, but the time frame is nevertheless short for making strong statements about consequences. Additionally, a longer time frame may be useful for understanding if differences between intrinsic and extrinsic motivational drivers may have, in the long-run, different effects concerning well-being and productivity. Previous research on workaholism has found that workers who are driven only by external motivation can be exposed to a progressive exhaustion of resources. For example, perfectionism, conscientiousness, and self-efficacy were associated with an increase in workaholism only in contexts where the workload was very high or the overwork climate was strong [73,74]. Thus, the apparent benefit of extrinsic work value orientation for employees’ well-being during presenteeism, as we found in the present study, requires more refined and nuance probing in future studies, ideally with a longer timeframe.

## 6. Conclusions

As argued by Cooper and Lu in addition to Ruhle et al., sickness presenteeism as a work behavior should be viewed as neutral; it is through the interaction between a person (as a motivational/regulatory being) and the environment (with resources and constraints) that a process for actualizing both the potential costs and benefits of such work behavior is set in motion [4,10]. In other words, the often negatively construed presenteeism behavior should be re-examined in contexts across multiple levels: individual, group, leader, organization, and overarching/social context [4]. We have taken the pioneering step to investigate the outcomes of sickness presenteeism from the above contextual perspectives. Our findings make substantial contributions to the existing knowledge on presenteeism by highlighting the underlying psychological mechanisms (i.e., motivational resources) in the integrated process of the “working long hours–presenteeism–outcomes” linkage. Specifically, intrinsic work value orientation alleviated the negative effect of long working hours on *job performance* via presenteeism, and extrinsic work value orientation alleviated the negative effect of long working hours on *well-being* via presenteeism. Our moderated mediation models also highlighted the largely overlooked temporal issue. The lasting effects of overworking on individual outcomes (well-being and performance) provide a more holistic understanding of the prevailing phenomenon of the intensification of work in today’s highly competitive, volatile, and uncertain business world.

## Figures and Tables

**Figure 1 ijerph-19-02179-f001:**
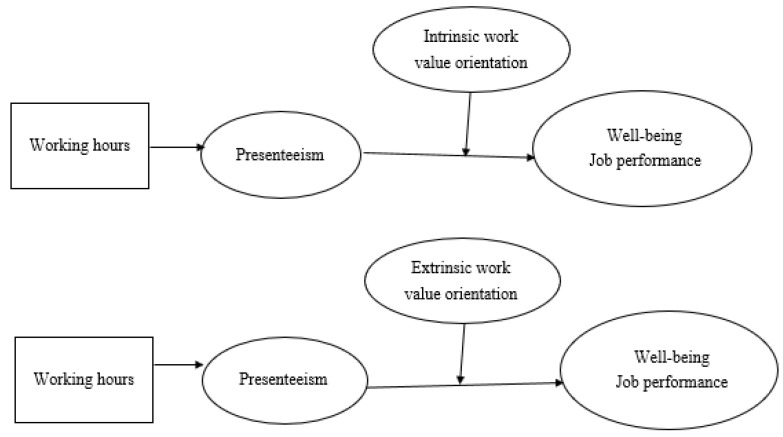
The research framework.

**Figure 2 ijerph-19-02179-f002:**
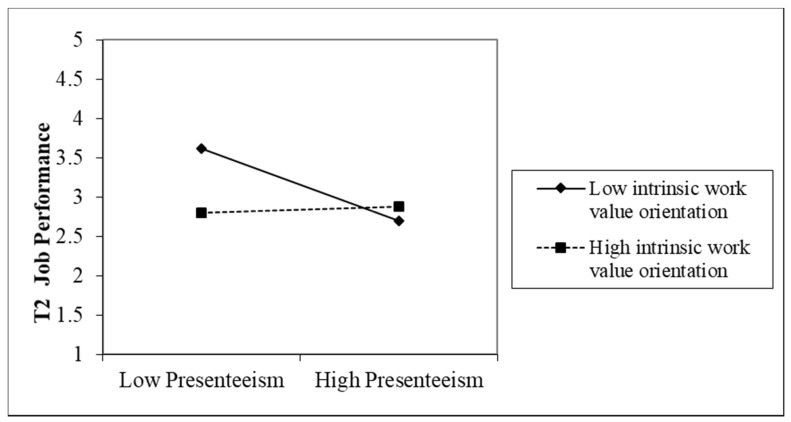
Interaction of presenteeism and intrinsic work value orientation: moderation on the “presenteeism–job performance” path.

**Figure 3 ijerph-19-02179-f003:**
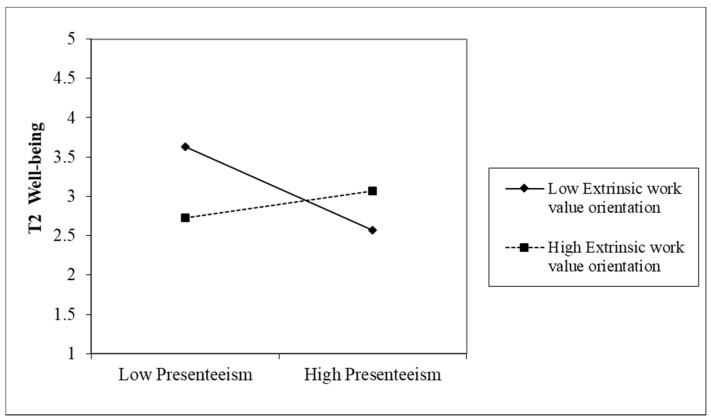
Interaction of presenteeism and extrinsic work value orientation: moderation on the “presenteeism–well-being” path.

**Table 1 ijerph-19-02179-t001:** Interrelations among research variables (N = 275).

	1	2	3	4	5	6	7	8	9	10	11	12
1. Sex	1											
2. Age	−0.09	1										
3. Marital status	−0.03	0.42 ***	1									
4. Job position	0.07	0.34 ***	0.12 **	1								
5. T1 working hours	0.11	0.03	−0.02	0.28 ***	1							
6. T1 sickness presenteeism	−0.16 **	0.10	0.04	0.03	0.17 ***	1						
7. T1 intrinsic work value orientation	0.12 *	0.06	−0.01	0.27 ***	0.21 ***	0.12 **	1					
8. T1 extrinsic work value orientation	0.01	0.08	−0.04	0.03	0.08	0.05	0.41 ***	1				
9. T1 job performance	0.05	0.21 **	−0.17 **	0.33 ***	0.26 ***	−0.14 **	0.35 ***	0.12 *	1			
10. T1 well−being	−0.03	0.15 **	−0.14 **	0.22 ***	−0.10 *	−0.16 **	0.34 ***	0.08 *	0.31 ***	1		
11. T2 job performance	0.05	0.12 **	−0.13 **	0.32 ***	−0.22 ***	−0.19 **	0.36 ***	0.1 **	0.66 ***	0.35 ***	1	
12. T2 well−being	0.12 *	0.21 **	−0.14 **	0.08	−0.13 **	−0.13 *	0.31 ***	0.12 *	0.25 ***	0.58 ***	0.40 ***	1
Mean	0.37	36.44	0.50	0.26	44.54	4.92	27.63	20.92	19.84	8.89	19.85	8.90
SD	0.47	8.34	0.48	0.43	7.36	1.79	3.87	2.51	3.29	2.42	3.37	2.48

Notes: sex: 0 = female, 1 = male; marital status: 0 = not married, 1 = married; and job position: 0 = employees, 1 = managers. * *p* < 0.05, ** *p* < 0.01, and *** *p* < 0.001.

**Table 2 ijerph-19-02179-t002:** Comparison of alternative factor structures for measurement validation.

Measurement Model	χ2	df	GFI	CFI	RMSEA
Time 1 variables					
Hypothesized three-factor model ^a^	1148.28	55	0.91	0.93	0.09
Two-factor model ^b^	1487.28	55	0.81	0.81	0.16
One-factor model ^c^	1487.28	55	0.74	0.63	0.21
Time 2 variables					
Hypothesized two-factor model ^d^	1378.80	36	0.95	0.97	0.08
One-factor model ^c^	1378.80	36	0.67	0.68	0.25

Notes: ^a^ Three-factor model (for time 1 variables): intrinsic work value orientation, extrinsic work value orientation, and presenteeism. ^b^ Two-factor model (for time 1 variables): work value orientations combined as factor 1; presenteeism as factor 2. ^c^ One-factor model (for time 1 or time 2 variables): all items loaded on one factor. ^d^ Two-factor model (for time 2 variables): well-being and job performance.

**Table 3 ijerph-19-02179-t003:** Bootstrapping effects and 95% confidence intervals (CI) for the mediation models.

Model Pathways	β (se)	*p*	95% CI (BCLL, BCUL)
Working hours → presenteeism	−0.137 (0.05)	0.001	(0.1358, 0.2599)
Working hours → well-being	−0.102 (0.05)	0.005	(−0.1049, −0.1033)
Working hours → job performance	−0.134 (0.04)	0.001	(−0.2321, −0.1576)
Working hours→ presenteeism → well-being	−0.140 (0.04)	0.005	(−0.1013, −0.0018)
Working hours→ presenteeism → job performance	−0.2110(0.03)	0.001	(−0.1207, −0.0110)

**Table 4 ijerph-19-02179-t004:** Model fit index and the index of moderated mediation models.

	TLI	CFI	GFI	RMSEA	β (se)	*p*	95% CI (BCLL, BCUL)
Base model one (well-being)	0.88	0.89	0.90	0.07	--	--	--
Base model two (job performance)	0.90	0.90	0.91	0.06	--	--	--
Model three (SPIN, well-being)	0.90	0.92	0.92	0.06	−0.011 (0.05)	0.132	(−0.0002, 0.0105)
Model four (SPIN, job performance)	0.92	0.93	0.94	0.05	−0.138 (0.03)	0.001	(−0.1238, −0.0052)
Model five (SPEX, well-being)	0.91	0.93	0.93	0.06	−0.162 (0.03)	0.001	(−0.2221, −0.1615)
Model six (SPEX, job performance)	0.90	0.91	0.92	0.06	−0.008 (0.07)	0.128	(−0.0102, 0.0003)

Notes: SPIN = latent interaction of sickness presenteeism × intrinsic work value orientation. SPEX = latent interaction of sickness presenteeism × extrinsic work value orientation. β = standardized estimate; se = estimated standard error; *p* = *p*-value; BCLL = lower limit of bias-corrected bootstrap confidence interval; BCUL = upper limit of bias-corrected bootstrap confidence interval. Model specification: Base model one = simple mediation model for well-being: working hours→presenteeism→well-being. Base model two = simple mediation model for job performance: working hours→presenteeism→job performance. Model three = moderated mediation model for well-being with intrinsic work value orientation as moderator. Model four = moderated mediation model for job performance with intrinsic work value orientation as moderator. Model five = moderated mediation model for well-being with extrinsic work value orientation as moderator. Model six = moderated mediation model for job performance with extrinsic work value orientation as moderator.

## Data Availability

The data that support the findings of this study are available from the corresponding author, L. Lu, upon reasonable request.

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
