# Peer review of "Sickness Presenteeism as a Link between Long Working Hours and Employees’ Outcomes: Intrinsic and Extrinsic Motivators as Resources"

_ijerph, 2022, doi:10.3390/ijerph19042179_

Round 1

Reviewer 1 Report

Earlier research suggests that presenteeism – going to work in spite of feeling too ill to work – has negative implications both for the individual (deteriorated health, burnout etc.) and for the organizations (lower productivity, contagious disease may spread at the work place, etc.).  This study problematizes the proposition that presenteeism will, unequivocally, have negative consequences for employees’ job performance and well-being. Using empirical materials from the Chinese island Taiwan, it tests three hypotheses – that long working hours are negatively associated with well-being and job performance (H1a and b), that this association is mediated via presenteeism (H2a and b), and that the linkages from long working hours to frequent presenteeism to less well-being/worse job performance are moderated by intrinsic and extrinsic work value orientations (H3a and b). The study utilizes two-wave panel data collected 5 months apart, with answers from 275 employees to questions in written questionnaires. Data are analyzed by means of “latent moderated structural equations”. Findings are in line with H1 and H2, and also H3 is at least partially supported, but not always (non-significant results in Model 3 and Model 6, Table 4). One of the authors’ conclusions is that organizations and managers should try to avoid too long working hours (in these data, the average working week was ca. 45 hours) in order to reduce presenteeism. Managers should furthermore accept that employees take sick leave when needed, and they should attempt to create work environments that foster both intrinsic and extrinsic work value orientations in order to minimize potential negative effects when presenteeism occurs.

In the opinion of this reviewer, the study has a number of qualities. The paper is well structured. It addresses a topic (presenteeism) which has attracted much attention in recent decades, for instance by health and managerial researchers. The “Theoretical framework and hypotheses” part is richly grounded in various psychological concepts and in previous research. The data are useful (but see below). Figure 1 illustrates the main idea well. The statistical analyses seem competently performed (this reviewer is no expert on SEM). Conclusions are basically in line with findings. The general finding is noteworthy: employees with high work value orientations appear to ‘suffer’ minimally from having many episodes of presenteeism (in terms of self-reported deterioration in well-being and job performance after 5 months).

              Thus, this reviewer has no weighty objections to this study, but will nevertheless add some remarks.

              The section on theoretical framework is complex reading, with many references, many concepts and detailed arguments. This reviewer had the feeling that some simplification could be made, in order to increase readability, without serious loss of content.

              As to the study’s originality, this reviewer thinks that the main hypothesis, that employees’ relationship to their work (high identification with organizational goals, being appreciated by clients and customers, etc.) reduces risk of negative consequences of presenteeism, is both quite plausible and also indicated in some previous studies. The authors claim that outcomes of presenteeism have been very sparsely studied (p.11-12, first para in Discussion), but is this an underrating? Early in the Discussion, it is claimed that the study has investigated “the entirety of the ‘antecedents-presenteeism-outcomes’ process”, which is a quite strong statement – too strong?

              As to limitations, this reviewer thinks the study would stand more firmly if a larger and more random sample of employees had been available. Measuring effects after 5 months is of course better than a simple cross-sectional study, but the time frame is nevertheless rather short for making strong statements about consequences of presenteeism. The authors are right in underlining the need for studying presenteeism in context, and they mention that “lack of firm-level controls may be an omitted variable bias”. Previous studies on presenteeism have highlighted the relevance of occupations, for instance in terms of the personal obligations teachers and care workers feel towards pupils and care recipients. The combination of a relatively small sample, short time frame, and little information about work places, respondents’ occupations, etc. are potential weaknesses – but this reviewer does not believe that this undermines the relevance of the study.

              Proof reading is recommended. It seems that a few sentences could be more clearly written. “Principle researcher” p.6 – should be “principal researcher”? Section on Measures, about Presenteeism: “four-point scales were used (0 = Never, 6 = More than five times)” – four-point and not seven-point scales? There are many unexplained abbreviations in the Results section – many of them may be familiar to those who know SEM (structural equation modelling), but some could perhaps be explained/written in full?

Author Response

Thank you very much for giving us the chance to improve the manuscript entitled “Sickness presenteeism as a link between long working hours and employees’ outcomes: Intrinsic and extrinsic motivators as resources” to ijerph. I have now incorporated suggestions made by the three reviewers in this round of review (Round 1). Below I will provide a point-by-point response and all changes have been made in the manuscript using the “Track Changes” function for easy checking.

Revisions made following Reviewer 1’s suggestions

1.…The section on theoretical framework is complex reading, with many references, many concepts and detailed arguments. This reviewer had the feeling that some simplification could be made, in order to increase readability, without serious loss of content.

Response: Thank you for the kind encouragement and suggestion. I have now shorten and simplified the “Theoretical Framework and Hypotheses Development” section. I have left out unnecessary references and repetitive arguments, while retaining the important conceptual content. Please note that the introduction to the motivational theory of excessive availability for work has been moved forward from the “Discussion” section, and several recent references have been added, following Reviewer 3’s specific suggestions.

  1. As to the study’s originality, this reviewer thinks that the main hypothesis, that employees’ relationship to their work (high identification with organizational goals, being appreciated by clients and customers, etc.) reduces risk of negative consequences of presenteeism, is both quite plausible and also indicated in some previous studies. The authors claim that outcomes of presenteeism have been very sparsely studied (p.11-12, first para in Discussion), but is this an underrating? Early in the Discussion, it is claimed that the study has investigated “the entirety of the ‘antecedents-presenteeism-outcomes’ process”, which is a quite strong statement – too strong?

Response: Thank you for the kind reminder. These statements have now been tune down and hopefully sound modest.

  1. As to limitations, this reviewer thinks the study would stand more firmly if a larger and more random sample of employees had been available. Measuring effects after 5 months is of course better than a simple cross-sectional study, but the time frame is nevertheless rather short for making strong statements about consequences of presenteeism. The authors are right in underlining the need for studying presenteeism in context, and they mention that “lack of firm-level controls may be an omitted variable bias”. Previous studies on presenteeism have highlighted the relevance of occupations, for instance in terms of the personal obligations teachers and care workers feel towards pupils and care recipients. The combination of a relatively small sample, short time frame, and little information about work places, respondents’ occupations, etc. are potential weaknesses – but this reviewer does not believe that this undermines the relevance of the study.

Response: Thank you for the pointy observation. I have now incorporated your comments and suggestions in “Limitations and Directions for Future Research” section. I have highlighted the relevance of occupations (para. 2) and need for a longer timeframe (para. 3).

  1. Proof reading is recommended. It seems that a few sentences could be more clearly written. “Principle researcher” p.6 – should be “principal researcher”? Section on Measures, about Presenteeism: “four-point scales were used (0 = Never, 6 = More than five times)” – four-point and not seven-point scales? There are many unexplained abbreviations in the Results section – many of them may be familiar to those who know SEM (structural equation modelling), but some could perhaps be explained/written in full?

Response: Thank you. I have made corrections to the typos. Presenteeism was measured with the four-point scales (1 = Never, 4 = More than five times), I have rectified the errors in scale markers. I have explained/written in full the technical abbreviations in the “Results” section, when they first appear.

Reviewer 2 Report

We believe the paper is within the scope of the journal, it brings empirical data that contributes to the discussion on the effects of long working hours to sickness presenteeism behavior and its outcomes on employee’s wellbeing and job performance, an important topic both for labor and companies. The writing is adequate, concise, and interesting. The title, abstract, and main sections of the paper are consistent with data presented. The aims are clearly stated and the method appropriate, with the benefit of employing a two-wave study design. The statistical analysis is suitable for the hypothesis testing, a model for the relationships between variables is provided. The conclusions are supported by data presented, figure and tables are informative. The references are extensive and adequate.

We have only a few small suggestions to authors:

Page 7, line 9: there is no reference to the percentage of respondents in medium enterprises, you should report that;

Page 7, line 28: you say the scale has “four points” but then you write that it goes from 0 to 6;

Page 7, line 31: when reporting the reliability of the scale (.88) you missed the alpha Cronbach symbol, that you use a few lines below, the reporting should be consistent;

Page 8, lines 3 and 4: when you say that “Preliminary analysis showed that social desirability score had no significant correlation with any of the main study variables” please state, in a few sentences, what method did you used;

Page 8, line 10: you missed the word “with” in the sentence “negatively correlated job performance and well-being”.

In our view, after small corrections, this paper is ready for publishing.

Author Response

Thank you very much for giving us the chance to improve the manuscript entitled “Sickness presenteeism as a link between long working hours and employees’ outcomes: Intrinsic and extrinsic motivators as resources” to ijerph. I have now incorporated suggestions made by the three reviewers in this round of review (Round 1). Below I will provide a point-by-point response and all changes have been made in the manuscript using the “Track Changes” function for easy checking.

Revisions made following Reviewer 2’s suggestions

…We have only a few small suggestions to authors:

  1. Page 7, line 9: there is no reference to the percentage of respondents in medium enterprises, you should report that;

Response: I have added the percentage of respondents in medium enterprises (27.5%).

  1. Page 7, line 28: you say the scale has “four points” but then you write that it goes from 0 to 6;

Response: Presenteeism was measured with the four-point scales (1 = Never, 4 = More than five times), I have rectified the errors in scale markers.

  1. Page 7, line 31: when reporting the reliability of the scale (.88) you missed the alpha Cronbach symbol, that you use a few lines below, the reporting should be consistent;

Response: I have added the alpha Cronbach symbol when reporting the reliability of the scale, to be consistent throughout.

  1. Page 8, lines 3 and 4: when you say that “Preliminary analysis showed that social desirability score had no significant correlation with any of the main study variables” please state, in a few sentences, what method did you used;

Response: I have a few sentences to explain the method we used to measure social desirability and the analyses we conducted to control for the social desirability bias (last para in “Method” section).

  1. Page 8, line 10: you missed the word “with” in the sentence “negatively correlated job performance and well-being”.

Response: Thank you. I have added the missing word.

Reviewer 3 Report

Dear authors,

I read have read your manuscript with interest. The study deals with an interesting topic and contributes to the development of the study on presenteeism.

I have some suggestion that I hope may improve the manuscript before publication.

The introduction is well made aand accurately introduce the topic. However I think that more recent literature on presenteeism should be cited, especially when the authors analyse the extant literature on the antecedents of presenteeism. For example:

Vinod Nair, A., McGregor, A. and Caputi, P. (2020), “The impact of challenge and hindrance demands on burnout, work engagement, and presenteeism. A cross-sectional study using the job demands–resources model”, Journal of Occupational and Environmental Medicine, Vol. 2 No. 8

Guidetti, G., Viotti, S., Converso, D., Sottimano, I. (2021) "Work and health-related factors of presenteeism: a mediation analysis on the role of menopausal symptoms between job demands and presenteeism among a sample of social service women employees",. International journal of workplace health management

McGregor, A., Magee, C.A., Caputi, P. and Iverson, D. (2016), “A job demands-resources approach to presenteeism”, Career Development International, Vol. 21, pp. 402-418, doi: 10.1108/CDI-01- 2016-0002

Moreover, in the introduction section, the authors declare that: "Taking the same motivational theoretical perspective, the above model views intrinsic and extrinsic work value orientations as motivational antecedents triggering differing types of presenteeis"

Can you better explain this statement, especially referring to "different types of presenteeism"? in this sense, could you better explain what kind of presentism conceptualization you intend to use in your study, as a loss of productivity or as the act of presenteesim due to ill health? This aspcct should be clearly defined since the introduction.

Moreover, I also suggest that the authors provide a better description of the working context where the study has been conducted, or the type of work (white collars) and to connect such description to the relevance of considering the specifc job demands, the long working hours, for their study.  This may introduce the reader to better understand the context and the development of hypotheses proposed by the authors.

Concerning the role of intrinsic and extrinsic work-related factors, authors may also more specifcally refer to the underlying theories and models. Indeed, they used such model within the discussion section, but in this version discussion and hypotheses development seems to be disconnected. Therefore, in order to let the reader to better understand, I suggest to introduce the models that the authors used to discuss the results and to corroborate their hypotheses (e.g motivational theory of excessive availability for work).

Concernign limits of the study, I think that a longer time frame may be useful to understand if differences between intrinsic and extrinc motivational drivers may have, on the long run different effect concerning wellbeing and productivity. That is to say, workers that are driven only by external motivation can be exposed to a progressive exhustion of resource. As it has been observed ( eg. Falco et al. 2020; Mazzetti et al., 2014) perfectionism, extrinsic motivation for achievement, narcissism are associated with an increase in workaholism only in contexts where the workload is very high or when the organizational climate favors high work commitment.

All these aspect may be considered for furure research development, that authors may improve.

The methods and materials  section is cleraly presented and methods used are appropriate.

The results may be improved in roder to let the reader better understand the results obtained. On the one hand by clearly highlight what paths emerged as significant and those who not, especially referring to table 4, a better descroption of the results may follow the table.

regarding this, I havce some concenrs regarding the model fits which are not totally satisfying. Therefore the authors should consider this limit and discuss it.

In my point of view the figure could be more clear: it is not clear on which path has been tested the moderation. Moreover, authors should better explain the steps that led to use latent moderators.

The discussion section is well made but can be improved regarding limits and future research developments.

Author Response

Thank you very much for giving us the chance to improve the manuscript entitled “Sickness presenteeism as a link between long working hours and employees’ outcomes: Intrinsic and extrinsic motivators as resources” to ijerph. I have now incorporated suggestions made by the three reviewers in this round of review (Round 1). Below I will provide a point-by-point response and all changes have been made in the manuscript using the “Track Changes” function for easy checking.

Revisions made following Reviewer 3’s suggestions

…I have some suggestion that I hope may improve the manuscript before publication.

  1. The introduction is well made and accurately introduce the topic. However I think that more recent literature on presenteeism should be cited, especially when the authors analyse the extant literature on the antecedents of presenteeism. For example:

Vinod Nair, A., McGregor, A. and Caputi, P. (2020), “The impact of challenge and hindrance demands on burnout, work engagement, and presenteeism. A cross-sectional study using the job demands–resources model”, Journal of Occupational and Environmental Medicine, Vol. 2 No. 8

Guidetti, G., Viotti, S., Converso, D., Sottimano, I. (2021) "Work and health-related factors of presenteeism: a mediation analysis on the role of menopausal symptoms between job demands and presenteeism among a sample of social service women employees". International journal of workplace health management

McGregor, A., Magee, C.A., Caputi, P. and Iverson, D. (2016), “A job demands-resources approach to presenteeism”, Career Development International, Vol. 21, pp. 402-418, doi: 10.1108/CDI-01- 2016-0002

Response: I have now incorporated your suggested recent references in the “Theoretical Framework and Hypotheses Development” section.

  1. Moreover, in the introduction section, the authors declare that: "Taking the same motivational theoretical perspective, the above model views intrinsic and extrinsic work value orientations as motivational antecedents triggering differing types of presenteeism." Can you better explain this statement, especially referring to "different types of presenteeism"? in this sense, could you better explain what kind of presentism conceptualization you intend to use in your study, as a loss of productivity or as the act of presenteesim due to ill health? This aspect should be clearly defined since the introduction.

Response: I have now stated outright that “The act of presenteeism due to ill health is the conceptualization we use in the present study” in the “Introduction” section (1st para). To avoid misunderstanding, I have re-written the sentence in question to read: “Taking the same motivational theoretical perspective, while Cooper and Lu’s model views intrinsic and extrinsic work value orientations as motivational antecedents manifested as different types of driving forces (i.e. voluntarily or involuntarily) for committing the presenteeism behavior, we in the present study attempted to unpack the mechanisms of work value orientations on the “presenteeism-outcomes” link” (3rd para).

  1. Moreover, I also suggest that the authors provide a better description of the working context where the study has been conducted, or the type of work (white collars) and to connect such description to the relevance of considering the specific job demands, the long working hours, for their study. This may introduce the reader to better understand the context and the development of hypotheses proposed by the authors.

Response: I have now added a few sentences describing the relevance of long working hours in the white-collar work In Taiwan (1st para in “The mediating role of presenteeism” subsection).

  1. Concerning the role of intrinsic and extrinsic work-related factors, authors may also more specifically refer to the underlying theories and models. Indeed, they used such model within the discussion section, but in this version discussion and hypotheses development seems to be disconnected. Therefore, in order to let the reader to better understand, I suggest to introduce the models that the authors used to discuss the results and to corroborate their hypotheses (e.g motivational theory of excessive availability for work).

Response: I have now moved forward the underlying theories and models (e.g., introduction to the motivational theory of excessive availability for work) from the “Discussion” section to the “Introduction” section (3rd para).

  1. Concerning limits of the study, I think that a longer time frame may be useful to understand if differences between intrinsic and extrinsic motivational drivers may have, on the long run different effect concerning wellbeing and productivity. That is to say, workers that are driven only by external motivation can be exposed to a progressive exhaustion of resource. As it has been observed (eg. Falco et al. 2020; Mazzetti et al., 2014) perfectionism, extrinsic motivation for achievement, narcissism are associated with an increase in workaholism only in contexts where the workload is very high or when the organizational climate favors high work commitment. All these aspect may be considered for future research development, that authors may improve.

Response: I have now incorporated your comments and suggested work in “Limitations and Directions for Future Research” section. I have highlighted the need for a longer timeframe to probe the effects of extrinsic work value orientation for employees’ well-being in future studies (para. 3).

  1. …The results may be improved in order to let the reader better understand the results obtained. On the one hand by clearly highlight what paths emerged as significant and those who not, especially referring to table 4, a better description of the results may follow the table. Regarding this, I have some concerns regarding the model fits which are not totally satisfying. Therefore the authors should consider this limit and discuss it.

Response: I have now revised the “Results” section better describe the procedures, analyses conducted and results obtained. I have redesigned Table 4 to make it more understandable, and linking better with the description of the procedure and results preceding and following the table. I have also clearly stated which moderated mediation effects were statistically significant and which were not: “Specifically, intrinsic work value orientation moderated the indirect effect of long working hours on job performance via presenteeism, whereas extrinsic work value orientation moderated the indirect effect of long working hours on well-being via presenteeism.”

  1. In my point of view the figure could be more clear: it is not clear on which path has been tested the moderation. Moreover, authors should better explain the steps that led to use latent moderators.

Response: I have redesigned Figure 1 to simplify the representation of the hypothesized models. I have reworded the caption of Figure 2 to make it clear on which path the moderation is tested.

  1. The discussion section is well made but can be improved regarding limits and future research developments.

Response: I have further elaborated on the limits and directions for future research developments.